# Models See Hallucinations: Evaluating the Factuality in Video Captioning

**Hui Liu** and **Xiaojun Wan**
Wangxuan Institute of Computer Technology, Peking University
{xinkeliuhui, wanxiaojun}@pku.edu.cn

## Abstract

Video captioning aims to describe events in a video with natural language. In recent years, many works have focused on improving captioning models' performance. However, like other text generation tasks, it risks introducing factual errors not supported by the input video. Factual errors can seriously affect the quality of the generated text, sometimes making it completely unusable. Although factual consistency has received much research attention in text-to-text tasks (e.g., summarization), it is less studied in vision-based text generation. In this work, we conduct the first human evaluation of the factuality in video captioning and annotate two factuality datasets. We find that 56% of the model-generated sentences have factual errors, indicating it is a severe problem in this field, but existing evaluation metrics show little correlation with human factuality annotation. We further propose a weakly-supervised, model-based factuality metric FactVC, which outperforms previous metrics on factuality evaluation of video captioning.[1]

## 1 Introduction

Video captioning is a challenging cross-modal task that aims to describe videos with natural language sentences. In recent years, video captioning has received much attention in computer vision and natural language processing communities. Substantial progress has been made to generate descriptions for videos that contain a single event (Venugopalan et al., 2015; Pan et al., 2020) or multiple events (Zhou et al., 2018b; Lei et al., 2020). However, like other text generation tasks, video captioning models risk introducing factual errors not supported by the input video. Examples are shown in Table 1. This paper defines factual errors (or hallucinations) as follows: a span of caption text that contradicts the video or describes something not appearing in

---

Video content:

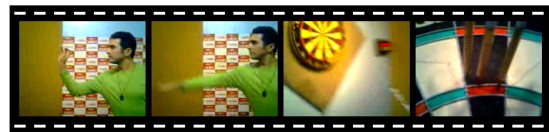

Caption 1:
A woman is throwing darts at a board.
She throws them at a board.
She jumps off into the distance and smiles.

Caption 2:
A man is seen standing in a room and leads into a man speaking to the camera.
The man is throwing darts at a dart board.
The man then throws the dart board and then goes back to the camera.

Caption 3:
A man in a white shirt is standing at a dart board.
He throws a dart at the end.

Table 1: The factual errors in video captioning. We show a video with three captions from different captioning models. Factual errors are marked in red.

---

the video. Factual errors can cause misunderstandings of the video content, sometimes making the generated captions completely unusable.

Factual consistency evaluation has received much research attention in text-to-text tasks, including summarization (Maynez et al., 2020; Kryściński et al., 2020), knowledge-grounded dialogue (Honovich et al., 2021), text simplification (Devaraj et al., 2022), and large language models(Bang et al., 2023). Nevertheless, it is less studied in vision-to-text tasks, especially video captioning. Therefore, this work focuses on the research gap in the factuality evaluation of video captioning.

Recently, more works have focused on videos with multiple events (Wang et al., 2021; Yamazaki et al., 2022), and it may bring more factual errors.

---

[1] Datasets and code will be released at https://github.com/PKULiuHui/FactVC.

So we choose multiple events video captioning for our evaluation. We use ActivityNet Captions (Krishna et al., 2017) and YouCook2 (Zhou et al., 2018a) as our video datasets, for they are the most common datasets for this task. Then we carefully select six recent models on each dataset to generate video captions. The models differ in model framework, pretrained features, and input signals. After collecting the videos and captions, we design a factuality annotation protocol and conduct human annotation. In the end, we obtain two human-annotated factuality datasets ActivityNet-Fact (200 videos, 3,834 sentences) and YouCook2-Fact (100 videos, 4,080 sentences). By analyzing the human annotation, we find that factual error (hallucination) is a severe problem in video captioning. To sum up, there are 87% of the paragraphs, 56% of the sentences, and 15% of the words have factual errors. There are different types of factual errors, including person-related errors, action-related errors, object-related errors and so on.

Since hallucination is a severe problem in video captioning, we test to what extent existing automatic evaluation metrics can measure the factuality of video captions. We find that most existing metrics correlate poorly with human judgment. Therefore we propose a new metric **FactVC** (**Fact**ual consistency for **V**ideo **C**aptioning). We use the CLIP model (Radford et al., 2021) to encode video frames and captions. Considering the CLIP model is trained on image-text pairs, it may have a gap transferring to video factuality evaluation. So we automatically construct a training set using text augmentation skills and finetune CLIP on it. Our FactVC metric achieves a higher correlation with human factuality annotation. The main contributions of this work are as follows:

- We conduct the first thorough factuality evaluation on video captioning. We find that hallucination is a severe problem while existing evaluation metrics can hardly measure it.

- We design a factuality annotation protocol and construct two human-annotated factuality datasets for video captioning.

- We propose a new factuality metric FactVC, which achieves a much higher correlation with human annotation on video captioning, and it can be further transferred to evaluate the factuality of image captioning.

## 2 Related Work

Factuality evaluation is first proposed in the field of document summarization. Maynez et al. (2020) conducted a human annotation on the XSUM dataset (Narayan et al., 2018) and found that more than 70% of summaries generated by summarization models have factual errors. Other human annotations(Wang et al., 2020; Pagnoni et al., 2021) reach similar conclusions. To measure the factual consistency, researchers proposed different metrics, which can be roughly divided into Entailment-based metrics (Falke et al., 2019; Kryściński et al., 2020) and QA-based metrics (Durmus et al., 2020; Wang et al., 2020; Scialom et al., 2021). Inspired by the works in summarization, factuality evaluation is studied for other tasks, including knowledge-grounded dialogue (Honovich et al., 2021), text simplification (Devaraj et al., 2022) and large language models (Bang et al., 2023).

For vision-based text generation tasks, the most widely used metrics are based on n-gram matching between references and generated captions, including BLEU(Papineni et al., 2002), ROUGE(Lin, 2004), METEOR(Banerjee and Lavie, 2005), CIDEr(Vedantam et al., 2015). They cannot match deeper semantics between captions and vision inputs. Recently, there are model-based metrics such as BERTScore(Zhang et al., 2019), CLIP-Score(Hessel et al., 2021), EMScore(Shi et al., 2022), PAC-S(Sarto et al., 2023). They leverage large-scale pretrained models to compute a matching score, even do not requiring reference captions.

A little work pays attention to the hallucination problem in vision-based text generation. CHAIR (Rohrbach et al., 2018) proposes an image relevance metric to evaluate the object hallucination in image captioning. They restrict their evaluation to 80 MSCOCO objects. COAHA(Ullah and Mohanta, 2022) uses object and action words matching to asses hallucination in video captioning. EMScore(Shi et al., 2022) and PAC-S(Sarto et al., 2023), designed for the overall evaluation of video captioning, also shows the potential to identify hallucinating captions. However, there is still no related human-annotated dataset and complete factuality evaluation work.

## 3 Human Annotation

Considering there does not exist factuality annotation of video captioning, we decide to construct our own datasets. We use ActivityNet Captions

(Krishna et al., 2017) and YouCook2 (Zhou et al., 2018a) as our source video datasets and select six recent models for each dataset to generate captions. Then we design a factuality annotation protocol and conduct our human annotation.

## 3.1 Datasets and Models

ActivityNet Captions(Krishna et al., 2017) contains 20k untrimmed videos of various human activities. Previous works(Lei et al., 2020; Yamazaki et al., 2022) report results on the ae-test split (2,457 videos). We randomly sample 200 videos from the ae-test split for human annotation. YouCook2 (Zhou et al., 2018a) contains 2,000 long untrimmed videos from 89 cooking recipes. Previous works report results on the val split (457 videos). We randomly sample 100 videos from the val split for human annotation.

We select six recent captioning models for each dataset and obtain the output captions on the sampled videos. For ActivityNet Captions, the selected models include: MART(Lei et al., 2020), COOT(Ging et al., 2020), PDVC-gt(Wang et al., 2021), PDVC-pred(Wang et al., 2021), Song(Song et al., 2021), VLTinT(Yamazaki et al., 2022). For YouCook2, the selected models include: VTrans(Zhou et al., 2018b), MART(Lei et al., 2020), COOT(Ging et al., 2020); COOT-100m(Ging et al., 2020), UniVL(Luo et al., 2020), VLTinT(Yamazaki et al., 2022). The above models are different in model framework, input signals, pretrained features, and pretraining scales.

## 3.2 Annotation Protocol

We design an annotation protocol to instruct annotators on how to measure and label the factuality of video captions. For the factuality annotation in summarization, annotators often give a binary label 0/1 for each summary sentence, indicating whether the sentence is factual or not (Maynez et al., 2020; Kryściński et al., 2020). However, the video captions have hierarchical structures (paragraph-sentence-word), and we want to obtain the factuality annotation for different granularity. So we design a new annotation protocol. Annotators are asked to give three levels of factuality annotation for each video caption. **Paragraph-level**: For each paragraph, annotators need to give a factuality Likert scale from 1 to 5, where 1 means the paragraph has many severe factual errors, and 5 means there are no obvious factual errors; **Sentence-level**: For each sentence, annotators give

a label 1 if it has factual errors else label 0; **Word-level**: Within each sentence, annotators need to mark phrases and words that have factual errors. It is worth noting that we focus on whether the caption has factual errors given the video (Precision) and do not care whether the caption describes the video completely (Recall). Refer to Appendix A for the complete annotation protocol and examples.

## 3.3 Annotation Procedure

According to previous works (Kryściński et al., 2020; Wang et al., 2020), the inter-annotator agreement through the crowdsourcing platform is relatively low. To make annotation more reliable, we hire three graduate students as our annotators. We provide them with a detailed instruction document and several annotation examples so that they can fully understand the annotation protocol. The annotations are checked multiple times during the annotation process. The annotations will be adopted only when an annotator completes all videos and passes every check. We collect three annotations for each video caption and combine them to get the final annotation. For paragraph-level annotation, we use the median score as the final score. For sentence-level and word-level annotation, we use the majority label as the final label. We quantified the degree of inter-annotator agreement using Krippendorff's alpha coefficient (Krippendorff, 2011). On the ActivityNet videos, the inter-annotator interval metrics are 0.750, 0.674, and 0.583 for paragraph-level, sentence-level, and word-level annotations respectively. On the YouCook2 videos, the inter-annotator interval metrics are 0.781, 0.774, 0.710 for paragraph-level, sentence-level, and word-level annotations respectively. The metrics show a substantial agreement between annotators. The agreement for word-level annotation is relatively low because it has more uncertainty and ambiguity.

## 4 Annotation Analysis

## 4.1 Datasets Statistics

Based on sampled ActivityNet and YouCook2 videos, we collect two annotated factuality datasets ActivityNet-Fact and YouCook2-Fact. The ActivityNet-Fact dataset contains 1,200 paragraphs, 3,834 sentences, and 48,235 words, among which 81.9% of the paragraphs, 51.4% of the sentences, and 13.5% of the words have factual errors. The YouCook2-Fact dataset contains 600 paragraphs,

| Category | Description | | Example | Ratio |
|---|---|---|---|---|
| Person | Person-related factual errors, e.g. gender, age, pronoun errors |  | A *woman* is throwing darts at a board. | 25.5% |
| Action | Action-related factual errors, not consistent with the video |  | The woman then begins *dancing* with the dog... | 38.1% |
| Object | Object-related factual errors, not consistent with the video |  | She then shows off a *rag* and speaking to the camera. | 19.9% |
| Adjective | Adjective-related factual errors, e.g. color, numerical errors |  | A person in a *red* shirt is walking towards the camera. | 6.6% |
| Poor Generation | Poor-generated sentence so that it contains factual errors |  | the bull is *UNK* and the bull is *UNK* . | 5.1% |
| Other | Other factual errors, e.g. relation errors, preposition errors |  | the person is riding the horses *in the air*. | 4.8% |

Table 2: Typology of factual errors in video captioning. Examples of each category are shown with a related video frame. Factual errors are marked in red italics. Ratios are shown on the ActivityNet-Fact dataset.

4,080 sentences, and 29,879 words, among which 98.3% of the paragraphs, 59.6% of the sentences, and 16.5% of the words have factual errors. This indicates that factual error is a severe problem in video captioning and should attract more research attention.

## 4.2 Factual Error Type Analysis

Factual errors have different categories and distributions in summarization task (Pagnoni et al., 2021). To analyze factual error types in video captioning, we conduct a post-analysis with our annotated datasets. We collect phrases/words marked as factual errors appearing at least twice and classify them into different error categories. The results are shown in Table 2. We can see that the factual errors in video captioning are various. For ActivityNet-Fact, the most common factual error categories are Person, Action, and Object, which count for 83.5% of the total factual errors. For YouCook2-Fact, Object is the dominant category, which counts for 92.3% of the total factual errors.

## 5 Metric Analysis

Now that factual errors broadly exist in video captions, we want to know to what extent existing metrics can measure the factuality of video captions. We test the correlation between automatic metrics and human annotation (for sentence/word-level annotation, we use the ratio of factual sentences and words as annotation score). We test model-free metrics BLEU(Papineni et al., 2002), ROUGE(Lin, 2004), METEOR(Banerjee and Lavie, 2005), CIDEr(Vedantam et al., 2015), COAHA(Ullah and Mohanta, 2022), and model-based metrics BERTScore(Zhang et al., 2019), EM-Score(Shi et al., 2022) [2]. The results are shown in Table 3 and Table 4.

| Metric | ref | Para | Sent | Word |
|---|---|---|---|---|
| Bleu4 | T | 0.178 | 0.173 | 0.174 |
| METEOR | T | 0.204 | 0.196 | 0.229 |
| Rouge-L | T | 0.170 | 0.151 | 0.185 |
| CIDEr | T | 0.151 | 0.141 | 0.133 |
| COAHA | T | 0.176 | 0.174 | 0.179 |
| BERTScore | T | 0.243 | 0.196 | 0.198 |
| EMScore | V | 0.305 | 0.242 | 0.341 |
| EMScore | T | 0.452 | 0.389 | 0.447 |
| EMScore | VT | 0.458 | 0.388 | 0.464 |

Table 3: Pearson correlation between automatic evaluation metrics and human annotation on the ActivityNet-Fact dataset. "ref" means the metric reference is human-written caption (T), input video (V), or both (VT).

To our surprise, the most commonly used met-

---

[2]For EMScore, we use the ViT-B/16 CLIP model, which performs better than the default ViT-B/32 CLIP model.

| Metric | ref | Para | Sent | Word |
|--------|-----|------|------|------|
| Bleu4 | T | 0.197 | 0.237 | 0.250 |
| METEOR | T | 0.411 | 0.371 | 0.415 |
| Rouge-L | T | 0.361 | 0.333 | 0.372 |
| CIDEr | T | 0.150 | 0.123 | 0.179 |
| COAHA | T | 0.193 | 0.236 | 0.286 |
| BERTScore | T | 0.426 | 0.403 | 0.434 |
| EMScore | V | 0.346 | 0.372 | 0.350 |
| EMScore | T | 0.524 | 0.501 | 0.537 |
| EMScore | VT | 0.543 | 0.530 | 0.555 |

Table 4: Pearson correlation between automatic evaluation metrics and human annotation on the YouCook2-Fact dataset.

rics for video captioning, such as Bleu4 and CIDEr, correlate poorly with factuality annotation. METEOR performs relatively better but still show a weak correlation with factuality annotation. As for model-based metrics, BERTScore shows little superior to METEOR on two datasets, indicating that just introducing large-scale text-pretrained model is not enough. EMScore leverages the image-text pretrained model CLIP(Radford et al., 2021) and shows a higher correlation with human annotation. In addition, it can evaluate video captions without human-written captions.

# 6 FactVC Metric

Although EMScore achieves a good correlation with human factuality annotation, it has two drawbacks: 1) EMScore uses the pretrained model CLIP, which is trained on image-text pairs from the Internet, and it may not transfer well to the video captioning data; 2) EMScore is designed for evaluating the overall quality of the video caption, not specifically designed for factuality evaluation. As a result, we propose a new metric **FactVC** (**Fact**ual consistency for **V**ideo **C**aptioning). We first automatically construct a factuality training set using text augmentation skills and then finetune the CLIP model. We also improve the calculation of the similarity score so that it is more suitable for factuality evaluation.

## 6.1 Training Data

Collecting a large-scale training dataset through human annotation is expensive and time-consuming. Inspired by (Kryściński et al., 2020; Gokhale et al., 2022), we decide to construct our training set au-

tomatically using text augmentation skills. Given a video $V$ together with a human-annotated caption sentence $T$, we use a set of text transformation functions to augment the dataset. The transformation functions include positive transformations ($\mathcal{T}^+$) which ensure the new sentence is factually correct and negative transformations ($\mathcal{T}^-$) which introduce factual errors into the sentence.

The positive transformations include: 1)Paraphrasing: we generate paraphrases using the back-translation method. We use the Google Translation API [3] and use German and French as middle language; 2) Simplification: we use a tool [4] to simplify complex and compound sentences into simple sentences. The negative transformations include: 1) Person Swap: we design a set of rules to change person words' gender, age, and pronoun; 2) Action Swap: we collect a common action set and apply deletion and insertion; 3) Object Swap: we collect a common object set and apply object substitution; 4) Adjective Swap: we swap adjectives (color, numerical words, etc.) in original sentences; 5) Poor Generation: we simulate the poor generation sentences by inserting "UNK" word and redundancy phrases. We design the negative transformations according to the factual errors shown in Table 2.

We first apply positive transformations to obtain fact-consistent sentences and then apply negative transformations to them to get sentences with factual errors. Finally, we collect a set of data samples $(V, T^+, T^-)$, where $V$ means the input video, $T^+$ means a fact-consistent sentence, and $T^-$ means the corresponding fact-inconsistent sentence. A detailed description of the data generation process is in Appendix B.

## 6.2 CLIP Finetuning

We only finetune the projection layers of the pretrained CLIP model. Given a batch of data $\{(V_i, T_i^+, T_i^-)\}_{i=1}^B$ with a batch size of $B$, we first use the CLIP model to calculate the similarities between each video and text:

$$s_{i,j}^+ = cos\left(\frac{1}{|V_i|}\sum_{k=1}^{|V_i|} E_v(f_{ik}), E_t(T_j^+)\right) \quad (1)$$

where $s_{i,j}^+$ means the similarity score between $V_i$ and $T_j^+$, $cos$ means cosine similarity, $f_{ik}$ is the

[3]https://translate.google.com/
[4]https://github.com/garain/Sentence-Simplification

$k$-th frame of video $V_i$, $|V_i|$ is the sampled frame number, $E_v$ and $E_t$ are the vision encoder and text encoder of CLIP model. The similarity score $s_{i,j}^-$ between $V_i$ and $T_i^-$ is computed similarly.

Then we finetune the CLIP model using the following loss function:

$$\mathcal{L}_{coarse} = -\sum_{i=1}^{B} \frac{exp(s_{i,i}^+)}{\sum_{j=1}^{B}(exp(s_{i,j}^+))} \quad (2)$$

$$\mathcal{L}_{fine} = \sum_{i=1}^{B} max(0, M - s_{i,i}^+ + s_{i,i}^-) \quad (3)$$

$$\mathcal{L} = \mathcal{L}_{coarse} + \lambda \mathcal{L}_{fine} \quad (4)$$

where $\mathcal{L}_{coarse}$ is a cross-entropy loss to learn whether the video content and text are matched, $\mathcal{L}_{fine}$ is a hinge loss to learn to assign a higher score to the fact-consistent text. $M$ and $\lambda$ are hyperparameters.

## 6.3 Score Calculation

With the finetuned CLIP model, we can calculate the factuality score FactVC as follows:

$$FactVC(V) = (1 - \alpha)S_c + \alpha S_f^p \quad (5)$$

$$S_c = cos(\frac{1}{|V|}\sum_{k=1}^{|V|} E_v(f_k), E_t(T)) \quad (6)$$

$$S_f^p = \frac{1}{|T|}\sum_{x_j \in T} \max_{f_i \in V} cos(E_v(f_i), E_t(x_j)) \quad (7)$$

where $\alpha$ is a balance factor, $S_c$ is the coarse-grained similarity score between video $V$ and sentence $T$. $S_f^p$ is the precision-based fine-grained similarity score computed between each frame $f$ and each word $x$.

Similar to EMScore, FactVC can use video $V$, human-written caption $T^*$, or both $(V, T^*)$ as reference. When using human-written caption $T^*$, we replace the video frame $f$ in Eq (6) and Eq (7) with the caption word $w$ and use the CLIP text encoder to compute $FactVC(T^*)$. $FactVC(V, T^*)$ is the average of $FactVC(V)$ and $FactVC(T^*)$.

We improve the calculation of FactVC metric in two aspects: 1) we use the precision-based score instead of the F-value-based score, for factuality is more related to the precision of video captions; 2) we introduce a parameter $\alpha$ to balance the coarse-grained score and the fine-grained score, we set it to 0.75 to favor more on fine-grained score.

## 7 Experiments

### 7.1 Comparison with Other Metrics

We compare the FactVC metric to other automatic metrics including Bleu(Papineni et al., 2002), ROUGE(Lin, 2004), METEOR(Banerjee and Lavie, 2005), CIDEr(Vedantam et al., 2015), COAHA(Ullah and Mohanta, 2022), BERTScore(Zhang et al., 2019), EMScore(Shi et al., 2022), PAC-S(Sarto et al., 2023). The results are shown in Table 5 and 6. We omit the metrics with worse correlation here (Bleu, ROUGE, METEOR, CIDEr, COAHA), and you can check them in Tables 3 and 4. From the table, EMScore and PAC-S perform better than BERTScore, indicating the usefulness of the CLIP model. However, they perform relatively poorly using video as the reference. Our FactVC metric, on the other hand, shows a much better performance in this setting. Compared to other metrics, FactVC shows the highest correlation with human annotation in all settings.

### 7.2 Ablation Study

We conduct an ablation study to test the effectiveness of each component in FactVC. The results are shown in Table 7. FactVC(no finetune) removes the finetuning process and shows an obvious performance degradation. FactVC($\mathcal{L}_{coarse}$) only uses $\mathcal{L}_{coarse}$ to finetune CLIP and FactVC($\mathcal{L}_{fine}$) only uses $\mathcal{L}_{fine}$ to finetune CLIP. From the results, we find that $\mathcal{L}_{coarse}$ can ensure stable finetuning, only using $\mathcal{L}_{fine}$ is not a good choice, but it can help video encoding together with $\mathcal{L}_{coarse}$. FactVC(F-value) uses the F-value-based score instead of the precision-based score, showing a performance degradation. This proves that factual consistency is more related to the precision of video captions. FactVC($\alpha = 0.5$) sets $\alpha$ to 0.5 in eq (5) and it is inferior to FactVC with $\alpha = 0.75$. This shows that the fine-grained score is more important in Factuality evaluation.

### 7.3 Model Ranking

Evaluation metrics are often reported at the system level to compare the performance of different models, and a reliable metric should be consistent with human judgment. We test the performance of the six models on the ActivityNet-Fact dataset using PAC-S and FactVC and report the average scores. The results are shown in Table 8. All the metrics are scaled to $[0, 1]$. Compared to human factuality annotation, PAC-S ranks the COOT, PDVC-gt,

| Metrics | Video as ref | | | Text as ref | | | Video & Text as ref | | |
|---|---|---|---|---|---|---|---|---|---|
| | Para | Sent | Word | Para | Sent | Word | Para | Sent | Word |
| BERTScore | - | - | - | 0.243 | 0.196 | 0.198 | - | - | - |
| EMScore (ViT-B/32) | 0.253 | 0.190 | 0.300 | 0.425 | 0.356 | 0.432 | 0.427 | 0.352 | 0.446 |
| EMScore (ViT-B/16) | 0.305 | 0.242 | 0.341 | 0.452 | 0.389 | 0.447 | 0.458 | 0.388 | 0.464 |
| PAC-S | 0.332 | 0.271 | 0.384 | 0.467 | 0.374 | 0.478 | 0.470 | 0.378 | 0.495 |
| FactVC | **0.462** | **0.371** | **0.480** | **0.511** | **0.438** | **0.498** | **0.551** | **0.465** | **0.545** |

Table 5: Pearson correlation between automatic metrics and human factuality annotation on the ActivityNet-Fact dataset. We test each metric in three settings: video as the reference, text as the reference, video and text as the reference. BERTScore only work with text as the reference. Metrics with worse correlation are omitted.

| Metrics | Video as ref | | | Text as ref | | | Video & Text as ref | | |
|---|---|---|---|---|---|---|---|---|---|
| | Para | Sent | Word | Para | Sent | Word | Para | Sent | Word |
| BERTScore | - | - | - | 0.426 | 0.403 | 0.434 | - | - | - |
| EMScore (ViT-B/32) | 0.337 | 0.353 | 0.361 | 0.518 | 0.482 | 0.523 | 0.543 | 0.514 | 0.553 |
| EMScore (ViT-B/16) | 0.346 | 0.372 | 0.350 | 0.524 | 0.501 | 0.537 | 0.543 | 0.530 | 0.555 |
| PAC-S | 0.312 | 0.335 | 0.331 | 0.543 | 0.516 | 0.546 | 0.562 | 0.544 | 0.570 |
| FactVC | **0.408** | **0.410** | **0.420** | **0.584** | **0.558** | **0.592** | **0.606** | **0.583** | **0.615** |

Table 6: Pearson correlation between automatic metrics and human factuality annotation on the YouCook2-Fact dataset. Metrics with worse correlation are omitted.

| Metrics | Video as ref | | | Text as ref | | | Video & Text as ref | | |
|---|---|---|---|---|---|---|---|---|---|
| | Para | Sent | Word | Para | Sent | Word | Para | Sent | Word |
| FactVC | **0.462** | **0.371** | **0.480** | **0.511** | **0.438** | **0.498** | **0.551** | **0.465** | **0.545** |
| FactVC(no finetune) | 0.349 | 0.281 | 0.388 | 0.497 | 0.425 | 0.483 | 0.512 | 0.433 | 0.510 |
| FactVC($\mathcal{L}_{coarse}$) | 0.427 | 0.348 | 0.466 | 0.508 | 0.435 | 0.492 | 0.537 | 0.456 | 0.532 |
| FactVC($\mathcal{L}_{fine}$) | 0.239 | 0.166 | 0.221 | 0.446 | 0.394 | 0.456 | 0.406 | 0.336 | 0.404 |
| FactVC(F-value) | 0.442 | 0.352 | 0.454 | 0.474 | 0.403 | 0.472 | 0.512 | 0.428 | 0.514 |
| FactVC($\alpha = 0.5$) | 0.444 | 0.366 | 0.458 | 0.485 | 0.423 | 0.474 | 0.523 | 0.450 | 0.518 |

Table 7: Ablation study on the ActivityNet-Fact dataset. The pearson correlation between each metric and human factuality annotation.

and PDVC-pred models differently. In contrast, FactVC ranks them consistently with human annotation.

### 7.4 Cross-Dataset Experiments

We conduct a cross-dataset experiment to test the generalizability of the FactVC metric. We use different datasets to finetune CLIP model and test them on ActivityNet-Fact and YouCook2-Fact datasets. The results are shown in Table 9. Compared to the CLIP model without finetuning, our finetuned method can obviously improve the metric

performance. Even training on a different dataset, the FactVC metric still performs well. Considering the huge domain gap between ActivityNet (ANet) and YouCook2 (You2) datasets, our FactVC metric has good generalizability on different video categories and textual styles.

### 7.5 Transferring to Image Captioning

According to the above experiments, FactVC performs well in evaluating the factuality of video captioning. We want to know whether our method can transfer to image captioning. So we ad-

| Models | Annotation | PAC-S | FactVC |
|---|---|---|---|
| MART | 0.413(6) | 0.602(6) | 0.569(6) |
| COOT | 0.446(4) | 0.649(3) | 0.615(4) |
| PDVC-gt | 0.553(2) | 0.647(4) | 0.633(2) |
| PDVC-pred | 0.513(3) | 0.652(2) | 0.631(3) |
| Song | 0.443(5) | 0.641(5) | 0.597(5) |
| VLTinT | 0.558(1) | 0.678(1) | 0.648(1) |

Table 8: Model performance with ranking on Sentence-level annotation and automatic metrics. We use video and text as the reference to compute PAC-S and FactVC metrics. The ranking of each model is shown in parentheses, and the rankings that are inconsistent with human annotation are marked in red.

| ActivityNet-Fact Dataset | | | | |
|---|---|---|---|---|
| Finetune Data | ref | Para | Sent | Word |
| None | | 0.512 | 0.433 | 0.510 |
| ANet | VT | 0.545 | 0.460 | 0.544 |
| You2 | | 0.540 | 0.457 | 0.533 |
| ANet + You2 | | **0.551** | **0.465** | **0.545** |
| YouCook2-Fact Dataset | | | | |
| None | | 0.572 | 0.555 | 0.587 |
| ANet | VT | 0.584 | 0.569 | 0.604 |
| You2 | | **0.611** | 0.582 | 0.613 |
| ANet + You2 | | 0.606 | **0.583** | **0.615** |

Table 9: FactVC metric cross-dataset experiments. We show the Pearson correlation between metrics and human annotation, using input video and human-written caption (VT) as reference.

| Metric | ref | Likert | Binary | Word |
|---|---|---|---|---|
| Bleu4 | T | 0.266 | 0.230 | 0.252 |
| METEOR | T | 0.308 | 0.239 | 0.295 |
| Rouge-L | T | 0.364 | 0.289 | 0.361 |
| CIDEr | T | 0.375 | 0.300 | 0.336 |
| CLIPScore | V | 0.359 | 0.220 | 0.297 |
| RefCLIPScore | VT | 0.457 | 0.298 | 0.398 |
| CLIPScore* | V | 0.364 | 0.238 | 0.309 |
| RefCLIPScore* | VT | 0.466 | 0.315 | 0.409 |
| CLIPScore** | V | 0.398 | 0.249 | 0.367 |
| RefCLIPScore** | VT | **0.513** | **0.341** | **0.478** |

Table 10: Pearson correlation between automatic metrics and human annotation on MSCOCO-Fact dataset. * and ** mean using video-finetuned and image-finetuned CLIP model respectively.

sults are shown in Table 10. CLIPScore and RefCLIPScore(Hessel et al., 2021) are CLIP-based metrics for image captioning. CLIPScore uses image as reference and RefCLIPScore additionally uses human captions as reference. CLIPScore* and RefCLIPScore* use our video-finetuned CLIP model. We also construct a training set from image caption data using the same text augmentation skills, and finetune the CLIP model to get CLIPScore** and RefCLIPScore**. From Table 10, we can see that with our finetuned method, CLIP-Score and RefCLIPScore can better measure the factuality of image captions.

For more implementation details, experiments and qualitative analysis, please refer to Appendix C and D.

# 8 Conclusion

In this work, we focus on the factuality evaluation in video captioning. We collect two human-annotated factuality datasets for video captioning and find that hallucination is a severe problem, with 56% of the model-generated sentences having different kinds of factual errors. However, most existing metrics show little correlation with human annotation. So we propose a new factuality metric FactVC. It is trained on an automatically-constructed training set and correlates much better with the factuality annotation. Experiments also show the potential of our method in evaluating image captions. Although factual consistency is a hot research topic in NLP tasks, it is less studied in video captioning. We hope our work can fill this

ditionally collect an annotated factuality dataset MSCOCO-Fact based on 200 MSCOCO(Lin et al., 2014) test images and five recent image captioning models' outputs. The selected models include: BUTD(Anderson et al., 2018), BUTD-sc(Anderson et al., 2018), VinVL(Zhang et al., 2021), OFA-base(Wang et al., 2022), OFA-huge(Wang et al., 2022). Unlike video captioning where a caption is a multi-sentence paragraph, an image caption is a single sentence. We collect three kinds of factuality annotation for each image caption: Likert (1-5 factuality score), Binary (0 or 1, indicating whether the sentence has a factual error), Word (whether each word has a factual error, and we use the ratio of factual words as word-level annotation score).

We test the correlation between image captioning metrics and human factuality annotation. Re-

research gap and promote further research in video captioning.

## Limitations

Due to time and financial constraints, we conduct our human annotation on the ActivityNet, YouCookII and MSCOCO dataset. In future, we hope to continue our evaluation on more different datasets (e.g. Charades Captions(Wang et al., 2018), MSR-VTT(Xu et al., 2016)).

## Ethics Statement

For human annotation, annotators are treated fairly and friendly. We paid them 12 dollars per hour, more than the local average minimum wage. We removed all content in the dataset that contains personal information about the annotators.

## Acknowledgments

This work was supported by National Key R&D Program of China (2021YFF0901502), National Science Foundation of China (No. 62161160339), State Key Laboratory of Media Convergence Production Technology and Systems and Key Laboratory of Science, Technology and Standard in Press Industry (Key Laboratory of Intelligent Press Media Technology). We appreciate the anonymous reviewers for their helpful comments. Xiaojun Wan is the corresponding author.

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

## A Annotation Protocol and examples

The detailed annotation instructions and protocol provided to the annotators are shown in Table 11. Two annotation examples are provided in Table 12.

## B Training Dataset Generation

A detailed description of the data generation process is shown in Algorithm 1.

## C Experiments

### C.1 Implementation details

We use the training split of ActivityNet Captions and YouCook2 to construct our training and validation set. The training and validation set size are

---

**Algorithm 1** The algorithm to generate dataset

**Require:**
  $S$ - set of videos $V$ and captions $T$
  $\mathcal{T}^+$ - set of positive transformations
  $\mathcal{T}^-$ - set of negative transformations

  **function** GENERATE_DATA($S, \mathcal{T}^+, \mathcal{T}^-$)
    $\mathcal{P} \leftarrow \varnothing$ ▷ set of positive data
    **for** $(V, T)$ **in** $S$ **do**
      $\mathcal{P} \leftarrow \mathcal{P} \cup \{(V, T)\}$
      **for** $fn$ **in** $\mathcal{T}^+$ **do**
        $T^+ \leftarrow fn(T)$
        $\mathcal{P} \leftarrow \mathcal{P} \cup \{(V, T^+)\}$
      **end for**
    **end for**
    $\mathcal{D} \leftarrow \varnothing$ ▷ set of data pairs
    **for** $(V, T^+)$ **in** $\mathcal{P}$ **do**
      **for** $fn$ **in** $\mathcal{T}^-$ **do**
        $T^- \leftarrow fn(T^+)$
        $\mathcal{D} \leftarrow \mathcal{D} \cup \{(V, T^+, T^-)\}$
      **end for**
    **end for**
  **end function**
  **return** $\mathcal{D}$

---

44,820 and 5,180 for ActivityNet and 18,029 and 1,971 for YouCook2. For CLIP finetuning, we start with the pretrained ViT-B/16 CLIP model. We sample three frames from each video clip uniformly. We set the margin $M$ in Eq (3) to 5.0 and the loss weight $\lambda$ in Eq (4) to 0.1. We finetune the projection layers of the CLIP model for three epochs with a batch size of 256 and learning rate of $5e-5$. During score calculation, we set the balance factor $\alpha$ in Eq (5) to 0.75, favoring fine-grained scores more than coarse-grained ones. Regarding the complexity cost, we finetune CLIP 3 epochs, which cost 4-6 hours on a single 2080Ti GPU card. We will keep the above settings unless otherwise stated.

### C.2 Extra ablation test

We explore the impact of the $\lambda$ value in Eq (4). The results are shown in Table 13. Note that when $\lambda = 0$, we only use $\mathcal{L}_{coarse}$ to finetune the CLIP model. From the table, FactVC performs better when choosing a $\lambda$ between $[0.1, 0.3]$. It makes the CLIP model make use of both $\mathcal{L}_{coarse}$ and $\mathcal{L}_{fine}$.

Considering FactVC is based on the image-text pretrained model CLIP(Radford et al., 2021), we want to explore the impact of using different CLIP models. The results are shown in Table 14. We

## Annotation Instructions

**About the task:**

Our task is video captioning, which uses AI models to generate text descriptions about the video content automatically. Currently, AI models often generate descriptions with factual errors that contradict the video or describe something not appearing in the video. Your task is to annotate the factual consistency between AI-generated captions and video content.

**About the annotation method:**

You need to annotate 200(100) videos in total. Each video contains six paragraph captions from different AI models, and each paragraph contains several sentences. You need to watch each video and then label the six captions. For each sentence, you need to judge whether it is fact-consistent with the video content. If the sentence is fact-inconsistent, mark the words/phrases with factual errors. For the whole paragraph, you need to give a factuality Likert scale(1-5, the higher, the better):

- 5: There are no obvious factual errors.
- 4: There are a few minor factual errors. Most parts are fact-consistent.
- 3: There are factual errors, but the fact-consistent contents are more.
- 2: There are more factual errors, and the fact-inconsistent contents are more.
- 1: There are a lot of severe factual errors. It can hardly describe the video content.

**Other tips:**

- AI models sometimes generate poor-quality descriptions, which may affect your annotation. If the sentence has minor grammar errors, you need to label it according to the corrected sentence. Otherwise, you need to label it as fact-inconsistent. There is a special word "UNK" (unknown word) which you should label as inconsistent.
- About the relationship between multiple sentences in a paragraph. The sentences are not in strict time order. The first sentence may describe the second half of the video, while the second may describe the first half. As a result, you should annotate each sentence separately and not consider the relation between multiple sentences.
- About the completeness of captions. You should only focus on the captions' correctness and not consider whether the caption describes the video completely. For example, caption 1 only describes a part of the video without factual errors, while caption 2 describes most video content with factual errors. The factuality of caption 1 is better than caption 2.
- About the commonsense. You can use commonsense during annotation. For example, if the video content is "a person laying on the bed with eyes closed", then the caption "a person is sleeping on the bed" is correct.
- About the annotation of phrases/words. You should mark as few words as possible if there are factual errors. For example, if the video content is about "A person lays on the bed", and the caption is "A person sits on the bed", you should mark "sits". If the video content is about "A person lays on the ground", and the caption is "A person sits on the bed", you should mark the whole phrase "sits on the bed".
- The order of AI models is shuffled. Do not assume the first caption comes from model 1, and the second caption comes from model 2, etc. After completing a video annotation, you should go back and check it and ensure your standard is consistent.
- We provide you with several annotation examples. Please read them before starting your annotation. This is very helpful for understanding the annotation method.

Table 11: The detailed annotation instructions and protocol that we provided to the annotators.

| Video content: | Video content: |
|---|---|
| 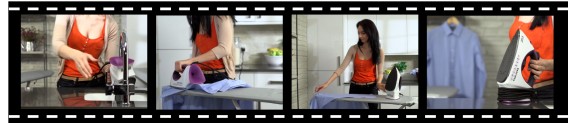 | 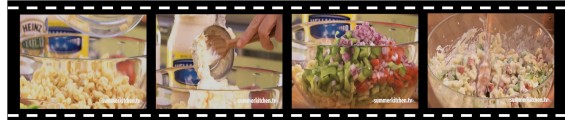 |

A man is seen speaking to the camera and leads into him holding up a pair of tools.

The man then begins ironing the shirt while speaking to the camera.

He continues to iron the iron and ends by showing off the finished product.

Paragraph score: 2

---

A woman is seen ironing a pair of pants on an ironing board while speaking to the camera.

She continues ironing the pants and ends by showing off the shirt.

Paragraph score: 2

---

A woman is ironing a shirt on an ironing board.

She shows off a pair of pants.

She then irons the shirt on the ironing board.

Paragraph score: 4

---

A woman is seen speaking to the camera and leads into a large iron of a large iron.

The woman then begins ironing the shirt and irons the iron.

The woman continues to iron the iron and shows off the iron.

Paragraph score: 3

---

She then shows the iron the iron and continues to use the iron.

She then irons the ironing the shirt and begins ironing the pants.

A woman is standing in a kitchen talking to the camera.

Paragraph score: 3

---

Add chickpeas lemon juice and lemon juice to a bowl.

Add flour salt pepper and a spoon of chicken breast and mix.

Add chopped tomatoes chopped spring onions and a little salt and pepper.

Toss and mix everything together.

Paragraph score: 2

---

Pour macaroni and milk on the pasta.

Add pasta sugar salt pepper and vinegar to the salad.

Add salt and pepper and mix.

Mix the salad.

Paragraph score: 4

---

Pour boiled macaroni and boiled macaroni in a bowl.

Add some mayonnaise and blend until smooth.

Add diced celery and minced garlic to a bowl.

Mix everything together.

Paragraph score: 4

---

Add 1 cup of chopped green onions and 1 cup of chopped green onions.

Plate the meat with the sauce and bread crumbs.

Add diced onion celery celery and mint to the food processor.

Mix the ingredients in the bowl.

Paragraph score: 2

---

Add pasta to a bowl.

Mix mayonnaise mayonnaise mayonnaise salt and pepper.

Add the cabbage celery and red bell pepper to the cabbage.

Toss the salad.

Paragraph score: 4

Table 12: Annotation examples. For each example, we show the video content, five paragraph captions, and the paragraph factuality scores. The phrases/words that are not factual are marked in red.

| FactVC λ value | Video as ref | | | Text as ref | | | Video & Text as ref | | |
|---|---|---|---|---|---|---|---|---|---|
| | Para | Sent | Word | Para | Sent | Word | Para | Sent | Word |
| $\lambda = 0.0$ | 0.421 | 0.343 | 0.448 | 0.508 | 0.435 | 0.490 | 0.537 | 0.455 | 0.528 |
| $\lambda = 0.1$ | **0.444** | **0.363** | 0.460 | **0.515** | 0.441 | **0.502** | **0.547** | **0.465** | **0.541** |
| $\lambda = 0.2$ | 0.438 | 0.355 | 0.462 | 0.514 | **0.443** | 0.499 | 0.543 | 0.461 | 0.536 |
| $\lambda = 0.3$ | 0.436 | 0.353 | **0.463** | 0.514 | **0.443** | 0.500 | 0.541 | 0.460 | 0.536 |
| $\lambda = 0.5$ | 0.427 | 0.346 | 0.459 | 0.513 | 0.442 | 0.500 | 0.537 | 0.457 | 0.534 |
| $\lambda = 1.0$ | 0.402 | 0.325 | 0.439 | 0.509 | 0.439 | 0.499 | 0.525 | 0.446 | 0.527 |

Table 13: Pearson correlation between FactVC(using different $\lambda$ value) and human factuality annotation on ActivityNet-Fact. We sample one frame from each video clip in this experiment. The best performance in each column is marked in bold.

| CLIP Model | Size | Video as ref | | | Text as ref | | | Video & Text as ref | | |
|---|---|---|---|---|---|---|---|---|---|---|
| | | Para | Sent | Word | Para | Sent | Word | Para | Sent | Word |
| RN50 | 102M | **0.317** | **0.265** | **0.357** | 0.478 | 0.412 | 0.480 | 0.488 | 0.418 | 0.497 |
| RN101 | 120M | 0.289 | 0.239 | 0.316 | 0.486 | 0.410 | 0.491 | 0.490 | 0.412 | 0.500 |
| RN50x4 | 178M | 0.282 | 0.230 | 0.328 | 0.478 | 0.403 | 0.497 | 0.480 | 0.403 | 0.507 |
| RN50x16 | 291M | 0.296 | 0.230 | 0.333 | **0.503** | **0.435** | **0.510** | **0.502** | **0.428** | **0.517** |
| RN50x64 | 623M | 0.276 | 0.216 | 0.320 | 0.470 | 0.401 | 0.466 | 0.467 | 0.394 | 0.474 |
| ViT-B/32 | 151M | 0.321 | 0.253 | 0.369 | 0.472 | 0.396 | 0.466 | 0.483 | 0.402 | 0.489 |
| ViT-B/16 | 150M | **0.349** | **0.281** | **0.388** | **0.497** | **0.425** | **0.483** | **0.512** | **0.433** | **0.510** |
| ViT-L/14 | 428M | 0.332 | 0.254 | 0.356 | 0.456 | 0.401 | 0.442 | 0.469 | 0.404 | 0.462 |
| ViT-L/14-336px | 428M | 0.340 | 0.268 | 0.364 | 0.454 | 0.400 | 0.441 | 0.468 | 0.405 | 0.463 |

Table 14: Pearson correlation between FactVC (using different CLIP models, no finetuning) and human factuality annotation on ActivityNet-Fact. We show each model's size (number of parameters). The best ResNet(RN) CLIP model and Vision-Transformer(ViT) CLIP model are marked in bold.

test FactVC performance on ActivityNet-Fact with different CLIP models without finetuning. The table shows that among ResNet-based CLIP models, RN50 and RN50x16 perform best; among ViT-based CLIP models, ViT-B/16 performs best. This leads us to the conclusion that a larger CLIP model does not necessarily perform better on factuality evaluation. As a result, we use the relatively small ViT-B/16 CLIP model in this work.

## C.3 Experiments on ActivityNet-FOIL

Shi et al. (2022) introduced the ActivityNet-FOIL dataset by automatically injecting foil visual concepts into the original captions from ActivityNet Captions ae-test split. It contains 1,900 correct-foil paragraph pairs, and at least one sentence in the foil paragraph contains a foil visual concept. This experiment uses different metrics to evaluate the correct-foil paragraph pairs and compute the pairwise ranking accuracy. The results are shown in

Table 15. We can see that when just using video as the reference, FactVC is better than previous metrics. When using both video and text as the reference, FactVC achieves the highest accuracy of 94.3%.

| Metric | Acc(%) | Metric | Acc(%) |
|---|---|---|---|
| BLEU1 | 60.1 | EMScore | 89.5 |
| BLEU4 | 66.1 | PAC-S | 90.1 |
| Rouge-L | 56.7 | FactVC | 91.0 |
| METEOR | 72.9 | EMScore* | 92.4 |
| CIDEr | 77.9 | PAC-S* | 93.5 |
| BERTScore | 86.7 | FactVC* | 94.3 |

Table 15: Pairwise ranking accuracy on ActivityNet-FOIL dataset. EMScore, PAC-S and FactVC use video as the reference. EMScore*, PAC-S* and FactVC* use both video and text as the reference.

Video 1:

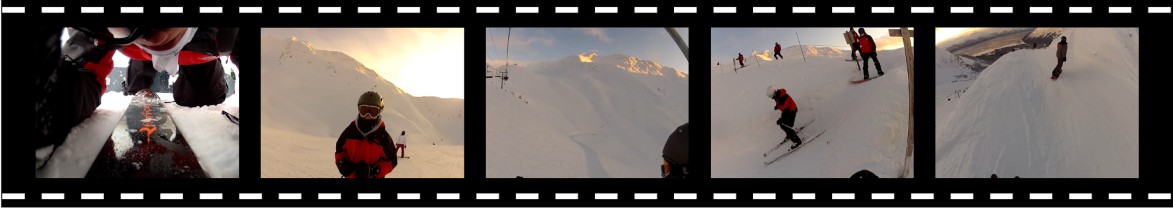

Generated caption:
A person is skiing down a hill of snow. They go over a hill of snow. They continue skiing down the hill together.

Reference captions:
A group of people are on a snowy mountain top. They are skiing down the numerous hills together. We see them flip and turn sharply in the driven snow.
A man is crouched down in the snow looking at the camera. He is then seen skiing through the snow. He is also seen riding the lifts before skiing again.

Paragraph: 1.0     Sentence: 1.0     Word: 1.0
Bleu2: 0.258     METEOR: 0.320     CIDEr: 0.238     BERTScore: 0.610
EMScore(V): 0.649     EMScore(VT): 0.805     FactVC(V): 0.838     FactVC(VT): 0.905

Video 2:

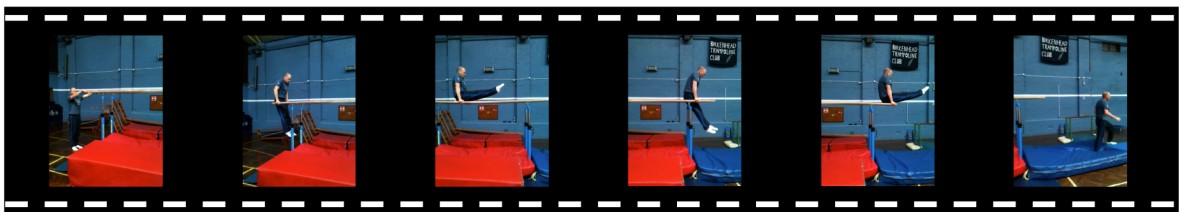

Generated caption:
He does a gymnastics routine on the bars. He does a gymnastics routine on the bars. He dismounts and lands on the bars.

Reference captions:
A man is seen standing before a set of uneven bars and begins inching himself forward. He raises his legs up when he stops and continues inches forward. He moves down all the way to end and jumps off into the mats in the end.
A man stands on front the parallel bars holding it. The man starts to advance holding on his hands. The man stops in the middle of the parallel bars, raise his legs and after continues advancing. Then, the man stops at the end of the bars, again he raises his legs, then exercises up and down. Next, the man jumps on the mat.

Paragraph: 0.75     Sentence: 0.667     Word: 0.957
Bleu2: 0.144     METEOR: 0.082     CIDEr: 0.0     BERTScore: 0.243
EMScore(V): 0.613     EMScore(VT): 0.582     FactVC(V): 0.711     FactVC(VT): 0.671

Table 16: Video captioning evaluation examples. For each example, we show the three-level factuality annotation scores and different metric scores. All scores are scaled in $[0, 1]$. For EMScore and FactVC, 'V' means using video as reference and 'VT' means using video and text as the reference. The factual errors are marked in red in the generated captions.

Video 3:

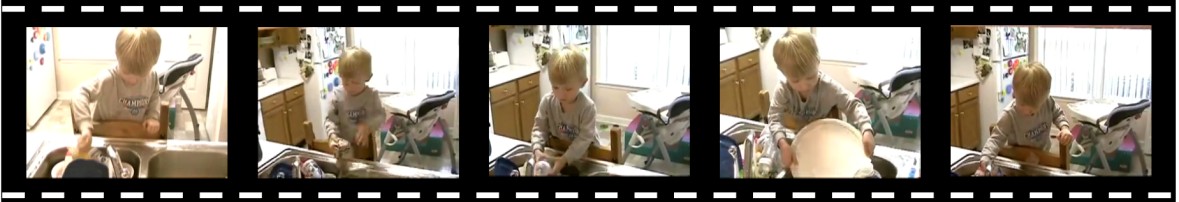

Generated caption:
A young boy is seen sitting behind a sink with a woman standing behind him. The boy then begins washing dishes while the boy watches from the side. The boy continues to brush his face and ends by turning off the camera.

Reference captions:
A toddler washes dishes in a sink while stand on a chair. The boy washes a cup, a sip cup and a dish. After, the boy jumps on the chair and then takes the dish again.
A small child is seen standing before a sink washing dishes. He wipes around the sink and continues washing dishes. He puts the clean dishes next to him.

Paragraph: 0.25     Sentence: 0.0     Word: 0.643
Bleu2: 0.379     METEOR: 0.448     CIDEr: 0.121     BERTScore: 0.724
EMScore(V): 0.723     EMScore(VT): 0.630     FactVC(V): 0.497     FactVC(VT): 0.522

Video 4:

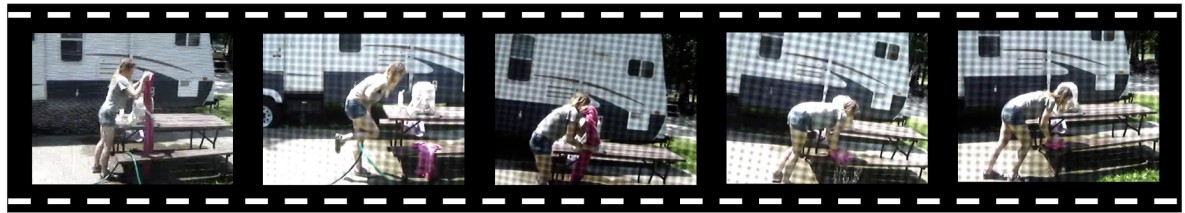

Generated caption:
A man is seen standing on a track with a stick. The man then begins playing with another person on a rope. The man continues playing and walks away.

Reference captions:
A woman is standing at a picnic table outside an RV. She is using water from a dispenser, pouring it onto material. She cleans the pink item with the water.
A lady is outside wringing a cloth on a bench. The lady places the pink cloth down on the bench. The lady removes a green hose from the brown bench. The lady press the cloth down on the bench and water drains.

Paragraph: 0.0     Sentence: 0.0     Word: 0.310
Bleu2: 0.180     METEOR: 0.072     CIDEr: 0.008     BERTScore: 0.373
EMScore(V): 0.152     EMScore(VT): 0.129     FactVC(V): 0.033     FactVC(VT): 0.072

Table 17: Continued with Table 16. Two more video captioning evaluation examples.

## D  FactVC Qualitative Analysis

We show several video captioning evaluation examples in Table 16 and 17. The annotation scores and metric scores are scaled in $[0, 1]$ for comparison. In video 1, the generated caption has no obvious factual errors. BERTScore, EMScore, and FactVC assign relatively high scores, while FactVC shows the most confidence. In video 2, the generated caption has a minor factual error, but it has a poor overlap with the references. All text-reference-based metrics give low scores, while FactVC gives a more reasonable factual score. In video 3, the generated caption has many different kinds of factual errors. However, probably because of the semantic overlap, BERTScore and EMScore give it high scores. Our FactVC gives a relatively low score. In video 4, the generated caption is full of severe factual errors, and FactVC correctly gives a very low score. The examples show that our FactVC metric performs best in measuring the factuality of video captions.