# OpenReview forum: "Models See Hallucinations: Evaluating the Factuality in Video Captioning"
_EMNLP/2023/Conference — EMNLP 2023 Main_

### Official Review · Reviewer_nfxG · 2023-07-30

**Soundness:** 5

**Excitement:**

3: Ambivalent: It has merits (e.g., it reports state-of-the-art results, the idea is nice), but there are key weaknesses (e.g., it describes incremental work), and it can significantly benefit from another round of revision. However, I won't object to accepting it if my co-reviewers champion it.

**Missing References:**

In the related work section, the paper should mention some relevant vision-based generation tasks and evaluation methods, such as (a) comprehensive analysis on multimodal generation [5], (b) a challenging image-based generation task: Visual storytelling [6,7,8,2,4], (c) reference-free metric for image-based generation [3], and (d) some other reference-free text-based metrics: [9].

-----
[1] UNION: An Unreferenced Metric for Evaluating Open-ended Story Generation, EMNLP 2020.

[2] No Metrics Are Perfect: Adversarial Reward Learning for Visual Storytelling, ACL 2018.

[3] Learning to Rank Visual Stories from Human Ranking Data, ACL 2022.

[4] Storytelling from an image stream using scene graphs, AAAI 2020.

[5] Towards Understanding Sample Variance in Visually Grounded Language Generation: Evaluations and Observations, EMNLP 2020.

[6] Visual storytelling, NAACL 2016.

[7] Show, Reward and Tell: Automatic Generation of Narrative Paragraph from Photo Stream by Adversarial Training, AAAI 2018.

[8] Plot and Rework: Modeling Storylines for Visual Storytelling, ACL 2021.

[9] BLEURT: Learning Robust Metrics for Text Generation, ACL 2020.

**Paper Topic And Main Contributions:**

This paper investigates factual error issues in video captioning. The authors begin by conducting human analysis and annotation to identify factual problems. They then propose a weakly-supervised factuality metric, FactVC, to address these issues. Extensive experiments show that FactVC outperforms other baselines, demonstrating its superiority.


**Questions For The Authors:**

(1) Have you considered using the original captions, without any positive transformation, as the positive samples?

(2) It is surprising that FactVC aligns exactly with the human ranking annotation in Table 8, as previous generation evaluation papers [2,3] have shown that ranking through many candidates (e.g., 6 in this table) can be very challenging. In such cases, model-generated texts should have equivalent qualities if they are all state-of-the-art models. I am wondering if this annotation was still conducted by the three annotators, and what is the inter-agreement of this analysis. If so, it might be a bit unconvincing for me. It would be ideal if the authors could provide some examples that include all the model-generated captions and human annotations.

(3) Have the authors considered pair-wise ranking for Table 8? Some papers [3,4] have shown that pair-wise ranking is more stable than ranking through many candidates at once, which might provide a more reliable evaluation for the model-generated captions.


**Reasons To Accept:**

(1) The paper is very well-written and easy to follow. The organization of the manuscript effectively follows their contributions.

(2) This paper's contributions, including both the dataset and the metric, will significantly benefit the development of video captioning.

(3) The authors conduct extensive experiments to demonstrate the improvements of this new evaluation method.

**Reasons To Reject:**

(1) Some relevant references are missing (see missing references section).

(2) The human analyses were conducted with relatively few annotators compared to other metric or evaluation papers, which could potentially lead to biased results due to the limited number of annotations. While Section 4 provides some fascinating analyses, the small number of annotations might make the conclusions less convincing. However, I personally find the analysis in Section 4.2 particularly intriguing, as it highlights the high error rates in recognizing objects and actions, indicating that current models still lack the ability to capture these crucial factors.

(3) The approach of creating positive and negative samples in this paper is similar to a previous study [1]. It would be beneficial to cite and discuss the distinctions between your work and their method in the paper.

**Reproducibility:**

3: Could reproduce the results with some difficulty. The settings of parameters are underspecified or subjectively determined; the training/evaluation data are not widely available.

**Reviewer Confidence:**

4: Quite sure. I tried to check the important points carefully. It's unlikely, though conceivable, that I missed something that should affect my ratings.

**Typos Grammar Style And Presentation Improvements:**

Missing space after 1) in line 334.

---

> ### Author Rebuttal · Authors · 2023-08-27
>
> **Thanks for your review! Below are our responses to your questions.**
>
> 1. **Have you considered using the original captions, without any positive transformation, as the positive samples?**
>
>    Yes. The direct way to collect positive samples is by using the original captions. However, the original captions' scale is small (36k for ActivityNet, 10k for YouCook2). In the initial research, we tried different positive and negative transformations. Positive transformations can enhance the diversity of samples and improve performance.
>
> 2. **Regarding the model ranking experiment in Table 8. How can the metric rank the models correctly? How to get the human annotation for model ranking?**
>
>    The chosen caption models are representative models in the last few years. Because they were proposed at different times (for now, the SOTA model is VLTinT), their performances are different. So a good metric can rank the models correctly.
>
>    We did not perform additional human annotation for the model ranking experiment. We directly use the human annotation in Section 3.3. We compute the average scores of captions from different models and use them to rank the models. All the annotators do not know which caption comes from which model.
>
> 3. **Have the authors considered pair-wise ranking for Table 8?**
>
>    When hiring annotators to compare different models, pairwise ranking can provide a more reliable evaluation. However, in our experiment, we did not hire annotators to rank the models. We use the annotation score in Section 3.3 and compute the average score to rank the models. A similar experiment is done in [1]
>
> 4. **Regarding the human annotation settings, why use relatively few annotators? Are the datasets relatively small?**
>
>    According to the previous work[2], factuality annotation through the crowdsourcing platform is not a good choice. We hire only three annotators so that we can train them well and check the annotations conveniently. Each model-generated caption is judged by three annotators, and we use the majority as the final annotation. Compared to factuality annotation in Text Summarization [2,3], the scale of our datasets is comparable or larger.
>
> 5. **Regarding the missing references, what are the distinctions between our text transformations and the transformations in [4]?**
>
>    Thank you for supplementing the references! Because our focus is video captioning (plus page limit), some work of other related tasks is not included. We will fully check the references and add the related work.
>
>    As for the distinctions between our work and [4], they are different in (1) we use paraphrasing and simplification as positive transformations while [4] use original data as positive samples; (2) our negative transformations are based on our analysis in Section 4.2, covering most factual error types while the negative transformations in [4] are designed for story generation (e.g., introducing grammar and logical errors.)
>
> **References**
>
> [1] EMScore: Evaluating Video Captioning via Coarse-Grained and Fine-Grained Embedding Matching. CVPR 2022
>
> [2] Evaluating the Factual Consistency of Abstractive Text Summarization. EMNLP 2020
>
> [3] On Faithfulness and Factuality in Abstractive Summarization. ACL 2020
>
> [4] UNION: An Unreferenced Metric for Evaluating Open-ended Story Generation, EMNLP 2020.

---

### Official Review · Reviewer_joFz · 2023-07-31

**Typos Grammar Style And Presentation Improvements:** None.
**Soundness:** 4

**Excitement:**

4: Strong: This paper deepens the understanding of some phenomenon or lowers the barriers to an existing research direction.

**Missing References:**

[1] vlcap: vision-language with contrastive learning for coherent video paragraph captioning. ICIP 2022.

**Paper Topic And Main Contributions:**

This paper first conducts the first human evaluation of the factuality in video captioning and annotate two factuality datasets. The authors find that 56% of the current model-generated sentences have factual errors. To this end, they further propose a new factuality metric FactVC for factuality evaluation of video captioning.



**Questions For The Authors:**

1.	Is there any reason to set the number of 200 and 100 for two newly human-annotated factuality datasets?
2.	Do the authors think about the possible solutions to address the problem of factual errors in video captioning?


**Reasons To Accept:**

1.	This paper is well-motivated, which aims to provide a thorough factuality evaluation for video captioning.
2.	This paper provides two human-annotated factuality datasets and a factuality metric for the task of video captioning, which could benefit the subsequent exploring of this task.
3.	The experiments are thorough.
4.	The writing is good and easy to follow.


**Reasons To Reject:**

1.	The scale of two human-annotated factuality datasets is relatively small, i.e., 200 videos and 100 videos.
2.	The used models of video paragraph captioning are relatively old, i.e., there is only one model proposed after 2022. The authors might considering test more recently designed models, such as VLCap [1].
3.	In the caption of Table 3: "ref" means the metric reference is human written caption (V), input video (V), or both (VT). I think that human written caption should be T.


**Reproducibility:**

4: Could mostly reproduce the results, but there may be some variation because of sample variance or minor variations in their interpretation of the protocol or method.

**Reviewer Confidence:**

4: Quite sure. I tried to check the important points carefully. It's unlikely, though conceivable, that I missed something that should affect my ratings.

---

> ### Author Rebuttal · Authors · 2023-08-27
>
> **Thanks for your review! Below are our responses to your questions.**
>
> 1. **Is there any reason to set the number of 200 and 100 for two newly human-annotated factuality datasets? Are the datasets relatively small?**
>
>    In this study, we use six models to generate paragraph-level captions for each video. Considering the number of models and caption length, annotation is time-consuming. We collect three annotations for every caption, further increasing the labor cost. As a result, we set the number of 200 and 100 for two datasets so that the scales of the two datasets are comparable, containing 3834 and 4080 sentences, respectively. Compared to factuality annotation in Text Summarization [1, 2], the scale of our datasets is comparable or larger.
>
> 2. **Possible solutions to address the problem of factual errors in video captioning?**
>
>    Reducing the factual errors in video captioning is challenging. Although the solution is beyond the topic of this paper, we can share some possible ways. First, we can adjust the training target of the caption model. For example, we can use contrastive learning or reinforcement learning skills to punish the generation of factual errors. Second, we can extract additional information from video content or external knowledge base and restrict the generated caption consistent with the information. We hope researchers propose more effective methods in the future.
>
> 3. **Are the used models of video paragraph captioning relatively old?**
>
>    We choose the representative models in the last few years. We did not choose VLCap[3] because we have chosen VLTinT[4]. They are done by the same authors and VLTinT[4] is an improved version of VLCap[3].
>
> 4. **Regarding the caption of Table 3, is there a mistake?**
>
>    Yes, we made a mistake in writing here. The human-written caption should be represented as T, as you pointed out. We will fix it in the revised version.
>
> **References**
>
> [1] On Faithfulness and Factuality in Abstractive Summarization. ACL 2020
>
> [2] Evaluating the Factual Consistency of Abstractive Text Summarization. EMNLP 2020
>
> [3] VLCap: Vision-Language with Contrastive Learning for Coherent Video Paragraph Captioning. ICIP 2022
>
> [4] VLTinT: Visual-Linguistic Transformer-in-Transformer for Coherent Video Paragraph Captioning. AAAI 2023

---

### Official Review · Reviewer_1XK7 · 2023-08-04

**Typos Grammar Style And Presentation Improvements:** 1. L138
**Soundness:** 4

**Ethical Concerns:**

Yes

**Excitement:**

4: Strong: This paper deepens the understanding of some phenomenon or lowers the barriers to an existing research direction.

**Missing References:**

N/A

**Paper Topic And Main Contributions:**

The paper constructs a dataset to analyze and measure factual errors in video captioning. Then, the authors propose FactVC, a weakly-supervised, model-based factuality metric. This metric is developed by finetuning the CLIP model, which is an image-based model. Experimental results demonstrate that FactVC achieves a higher correlation compared to other metrics.

**Questions For The Authors:**

1. While FactVC demonstrates better correlations in factuality, it is important to consider other evaluation standards such as accuracy, naturalness, and so on. In other words, can we solely rely on FactVC for evaluating the quality of captions?
2. In Table 7, why does FactVC (L_fine) exhibit a significant drop compared to L_coarse? (L433)
3. The paper states, "Similar to EMScore, FactVC can utilize video V, human-written caption T*, or both (V, T*) as reference" (L394-L396). However, the equations provided only demonstrate the usage of video V. How can we incorporate the usage of T* and (V, T*)?
4. In Section 7.3, the authors employ an average score and rank the models. However, would it be more appropriate to assign a separate rank to each caption and then calculate a rank correlation (such as Spearman's rho) across all captions?

**Reasons To Accept:**

1. Hallucination poses as a fundamental issue in LLM (or the entire deep learning-based field, broadly speaking). The paper specifically addresses factual errors in video captions, which, according to their annotations, prove to be a significant problem (L73).
2. FactVC is a more effective metric for detecting factual errors in captions.
3. The paper is well-organized and provides a clear explanation of the topic.

**Reasons To Reject:**

1. The definition of factual errors or hallucinations provided in the paper (L40-L42) appears to be vague and general. It seems that most caption errors can be classified as hallucinations since content words form the main part of the sentences. However, it is unclear what other types of errors exist if they are not factual. Grammar errors come to mind, but it is unlikely that they constitute the primary type of error in generated captions. Therefore, the high number of hallucinations raises some suspicion.
2. It seems that FactVC only involves fine-tuning CLIP from image to video, which, if true, may not be a significant contribution. Additionally, instead of using CLIP for videos, why not utilize VideoCLIP [1], which is specifically designed for zero-shot video-text understanding?

[1] "VideoCLIP: Contrastive Pre-training for Zero-shot Video-Text Understanding"

**Reproducibility:**

4: Could mostly reproduce the results, but there may be some variation because of sample variance or minor variations in their interpretation of the protocol or method.

**Reviewer Confidence:**

4: Quite sure. I tried to check the important points carefully. It's unlikely, though conceivable, that I missed something that should affect my ratings.

---

> ### Author Rebuttal · Authors · 2023-08-27
>
> **Thanks for your review! Below are our responses to your questions.**
>
> 1. **Can we solely rely on FactVC for evaluating the quality of captions?**
>
>    FactVC is designed and trained to evaluate the factuality of video captions. We did not test it for other evaluation standards. We recommend using FactVC and other metrics together to evaluate the overall quality of captions.
>
> 2. **Why does FactVC (L_fine) exhibit a significant drop compared to L_coarse in Table 7?**
>
>    L_coarse is similar to the original CLIP loss function that trains video-text matching. L_fine trains the model to assign a higher score to fact-consistent text. Only using L_fine to finetune the CLIP model will destroy the video-text matching ability, causing a performance drop.
>
> 3. **How to use T\* and (V, T\*) to compute FactVC score?**
>
>    When using human-written caption T\*, we replace the video frame $f$ in Eq (6) and Eq (7) with the caption word $w$ and use the CLIP text encoder to compute FactVC(T\*). FactVC(V, T\*) is the average of FactVC(V) and FactVC(T\*). We will improve the equations in the revised version.
>
> 4. **Would it be more appropriate to conduct a caption-level ranking experiment?**
>
>    Section 7.3 conducts a model-level (system-level) ranking experiment. We design this experiment mainly because researchers often use system-level metric scores to compare different models. Caption-level ranking is also a good supplemental experiment.
>
> 5. **Are there other types of errors other than factual errors in video caption?**
>
>    Video caption errors contain content-unrelated errors (e.g., grammar errors) and content-related errors (e.g., factual errors, content-missing errors). Factual errors are the main part of content-related errors. Besides, many video captions fall into the "right but not good" category. We treat these captions as fact-consistent in this study.
>
> 6. **Are FactVC only involves fine-tuning CLIP from image to video? Why not using video-text models like VideoCLIP?**
>
>    The training of FactVC is not simply fine-tuning CLIP from image to video. We use text augmentation skills to construct the training set and leverage different loss functions. Besides, we design a scoring method to make it more appropriate for factuality evaluation. The reason for using CLIP is that it can be used to evaluate video and image captions (Section 7.5). VideoCLIP may be suitable for video caption and needs experimental verification.

---

### Meta-Review · Area_Chair_ii5K · 2023-09-17

**Recommendation:** 5

**Metareview:**

This paper studies an interesting topic of hallucination in video captioning models. All reviewers positively rated the work and agreed that contributions of the paper, namely two datasets and a novel metric, are significant and exciting to the field. In addition, all reviewers also commented high on the quality of the paper.

---

### Decision · Program_Chairs · 2023-10-07

**Decision:**

Accept-Main

**Comment:**

This paper studies an interesting topic of hallucination in video captioning models. All reviewers positively rated the work and agreed that contributions of the paper, namely two datasets and a novel metric, are significant and exciting to the field. In addition, all reviewers also commented high on the quality of the paper.